# First Isolation, Antifungal Susceptibility, and Molecular Characterization of *Cryptococcus neoformans* from the Environment in Croatia

**DOI:** 10.3390/jof5040099

**Published:** 2019-10-12

**Authors:** Donjeta Pllana-Hajdari, Massimo Cogliati, Ljiljana Čičmak, Sanja Pleško, Emilija Mlinarić-Missoni, Ivana Mareković

**Affiliations:** 1Department of Molecular Microbiology, National Institute of Public Health, 10 000 Prishtina, Kosovo; 2Laboratorio di Micologia Medica, Dipartimento Scienze Biomediche per la Salute, Università degli Studi di Milano, 20133 Milano, Italy; Massimo.cogliati@unimi.it; 3Department for Parasitology and Mycology, Croatian Institute for Public Health, 10 000 Zagreb, Croatia; ljiljana.cicmak@hzjz.hr (L.Č.); emilija.mlinaric.missoni@gmail.com (E.M.-M.); 4Department of Clinical and Molecular Microbiology, University Hospital Centre Zagreb, University of Zagreb School of Medicine, 10 000 Zagreb, Croatia; sanja.plesko@kbc-zagreb.hr

**Keywords:** *Cryptococcus neoformans*, environment, molecular types, antifungal susceptibility

## Abstract

The purpose of this study was to investigate the presence of *Cryptococcus neoformans* species complex isolates from environmental sources in Croatia and to determine their molecular types and antifungal susceptibility. Swab samples of tree hollows and bird excreta in the soil beneath trees were collected. Samples included 472 (92.73%) samples obtained from tree hollows and 37 (7.27%) samples from bird excreta. Four *C. neoformans* species complex isolates were recovered from tree hollow swabs along the Mediterranean coast, while there were no isolates recovered from bird excreta or from the continental area. Three isolates were identified as molecular types VNI and one as VNIV. All tested antifungals showed high in vitro activity against the four isolates. This is the first report proving the presence of *C. neoformans* species complex in the environment of Croatia. The results of the study suggest a major risk of exposure for inhabitants living along the Croatian coast and that both VNI and VNIV molecular types can be expected in clinical cases of cryptococcosis. Susceptibility to antifungals confirmed that no resistance should be expected in patients with cryptococcosis at the present time.

## 1. Introduction

The encapsulated basidiomycetous yeasts of *Cryptococcus neoformans* and *C. gattii* species complexes are infectious agents of cryptococcosis, a life-threatening infection primarily affecting immunocompromised hosts [1]. While the *C. neoformans* species complex mainly affects patients with acquired immunodeficiency syndrome (AIDS) and those who are immunosuppressed (transplant patients, those on long-term corticosteroids, and those prescribed monoclonal antibodies), one-quarter of patients with *C. gattii* species complex infections are immunocompetent and healthy. Infection of the brain and meninges by the *C. neoformans* species complex is the most important clinical manifestation in immunosuppressed individuals [2,3]. An estimated 220,000 cases of cryptococcal meningitis complicate HIV/AIDS worldwide each year, resulting in nearly 181,000 deaths annually [3,4].

Infections by *Cryptococcus* species are acquired from environmental sources and are a consequence of the inhalation of dehydrated blastoconidia or basidiospores into the lungs. The yeasts have been isolated from bird excreta, soil, bark and trunk hollows of trees, and decaying wood in various parts of the world [5].

The use of antifungal agents, particularly in long-term suppressive regimens, has raised concerns about the development of drug resistance in *C. neoformans* species complex [6]. Globally, fluconazole-resistant strains of *C. neoformans* species complex have been increasingly reported in the past two decades. Geographical distribution shows that increasing fluconazole resistance is demonstrated in Africa, Asia, and Latin America, while low rates are still found in North America and Europe, except for Spain [6,7,8]. In Croatia, the clinical isolates investigated in one study showed no resistance either to fluconazole or other antifungals [9]. A few studies reporting the antifungal susceptibility of environmental *C. neoformans* species complex strains have been conducted, mainly in Brazil and India [10,11,12]. Such studies are significant because the susceptibility data of environmental isolates may influence the profiles of the clinical isolates recovered from patients because of the transmission pathway from environmental sources. However, such studies from Europe are lacking.

The taxonomy of *C. neoformans* is still under major investigation [13,14,15,16]. The term “species complex” is used to comprise all genetic, pathogenic, epidemiological, ecological, and clinical differences between the strains [14]. *C. neoformans* species complex currently consists of five major molecular types distinguishable by several molecular techniques [17,18,19,20,21]: VNI, VNII, and VNB, with capsular antigen A (serotype A) and classified as *C. neoformans* var. *grubii*, VNIV with capsular antigen D (serotype D) and classified as *C. neoformans* var. *neoformans*, and VNIII, including diploid and aneuploid hybrids between the two varieties (AD hybrids). Molecular epidemiology data are important because they enable us to understand the population structure of the pathogen, its evolution, and how it is spreading across continents. In Europe, most cases of cryptococcosis are caused by isolates of the *C. neoformans* species complex, with VNI being the most prevalent molecular type, followed by VNIV and VNIII [5,22]. Genotyping of 48 clinical isolates obtained from 15 patients in Croatia was performed by amplified fragment length polymorphism (AFLP), showing a prevalence of 40% AFLP1/VNI, 40% AFLP2/VNIV, and 20% AFLP3/VNIII isolates [9]. At present, *C. neoformans* species complex strains from environmental sources in Croatia have not yet been isolated or investigated for antifungal susceptibility and molecular epidemiology.

The purpose of this study was to investigate the presence of *C. neoformans* species complex isolates from environmental sources in Croatia and to determine their molecular types and antifungal susceptibility. The research results will provide the first insights into the ecology of *Cryptococcus* species in Croatia and thus the potential exposure risk of the inhabitants to these yeasts in the investigated urban locations. The antifungal susceptibility profile of *C. neoformans* isolated from environmental sources can indicate the susceptibility of clinical isolates and lead to the development of treatment guidelines.

## 2. Materials and Methods

### 2.1. Geographic and Climate Characteristics of Croatia

The geography of Croatia is defined by its location in southeastern Europe along the Mediterranean coast (Figure 1). Due to this location at the meeting point of the Mediterranean, the Alps, and the Pannonian plain, Croatia exhibits great geographical and natural diversity. The largest part of Croatia has a moderately warm rainy climate under the Köppen classification, with a mean monthly temperature in the coldest month of the year above ‒3 °C and below 18 °C [23]. 

### 2.2. Sample Collection, Cultures, and Identification

Swab samples of tree hollows and bird excreta in the soil beneath trees were collected from different geographical areas in Croatia. Trees involved were identified to the species level whenever possible. The samples were collected from urban public places including squares, parks, gardens, hospital areas, and school playgrounds. Collecting and processing of swab samples was performed as described elsewhere [24]. Samples were cultured onto plates containing Niger seed agar incubated at 27 °C and 37 °C for two weeks and were monitored daily. The dark brown colonies suggestive of cryptococcal colonies were subcultured on Sabouraud peptone dextrose agar plates (Liofilchem, Roseto degli Abruzzi, Italy). The number of brown colonies appearing on each culture plate was counted and recorded. Identification of *C. neoformans* by conventional mycological methods included the commercial identification system ID 32 C (BioMérieux, Marcy l’Etoile, France) based on the assimilation of carbohydrates and nitrate as well as the presence of capsule observed in India ink preparations. Up to 10 cryptococcal yeast colonies were stored in glycine buffer at −20 °C until antifungal susceptibility testing and molecular analysis were performed.

### 2.3. Antifungal Susceptibility Testing

Antifungal susceptibility of *C. neoformans* species complex isolates to flucytosine, amphotericin B, fluconazole, itraconazole and voriconazole were determined by ATB FUNGUS 3 (BioMérieux, Marcy l’Etoile, France). The test was performed according to the manufacturer’s instructions. Isolates were also tested with a standard broth microdilution method according to the Clinical Laboratory Standards Institute (CLSI) for amphotericin B, fluconazole, and voriconazole [25]. Since there are no clinical breakpoints defined by the CLSI for *C. neoformans* species complex, epidemiological cutoff values (ECV) were used as reference values to define a strain as wild type or non-wildtype [26,27]. Isolates were considered wild type when the minimal inhibitory concentration (MIC) value for amphotericin B was ≤0.5 μg/mL, for flucytosine ≤8 μg/mL, for fluconazole ≤8 μg/mL, for itraconazole ≤0.25 μg/mL, for voriconazole ≤0.25 μg/mL. Susceptibility testing according to CLSI was done in 2019, two years after other investigations were already finished. For that reason, not all isolates (but two) were viable and available for testing.

### 2.4. Molecular Analysis

The molecular type and mating type of the isolates was determined by multiplex PCR, as previously described [18,28,29]. Strains H99 (VNI), WM626 (VNII), JEC21 (VNIV), and CBS 132 (VNIII) were included as reference strains [19].

### 2.5. Statistical Analysis

Statistical analysis included descriptive frequency tables, and Fisher’s exact test. Statistical significance was determined when *p* ≤ 0.05 was observed.

## 3. Results

### 3.1. Distribution of Samples According to Location, Sample Type, and Tree Species

A total of 509 swab samples was collected, covering a wide territory of Croatia during summer and autumn 2013‒2016 (Figure 1). Fourteen Croatian towns were included in the study: Zagreb, Čakovec, Samobor, Varaždin, and Osijek in the continental area (*n* = 208, 40.86%), and Split, Rijeka, Rovinj, Šibenik, Korčula, Krk, Pula, Dubrovnik, and Makarska along the Mediterranean coast (*n* = 301, 59.13%). Samples included 472 (92.73%) swab samples from tree hollows and 37 (7.27%) samples from bird excreta. A total of 46 different tree species were sampled (Appendix A). Most of the trees were *Olea europea* (olive tree, 25.5%), *Ceratonia siliqua* (carob tree, 5.3%), *Pinus sylvestris* (scots pine, 5.1%), *Betula pendula* (silver birch, 4.5%), *Castanea* spp. (chestnut tree, 4.3%), and *Prunus dulcis* (almond tree, 3.9%) [30].

### 3.2. Cryptococcus *spp.* Isolates in Environmental Samples

Four *C. neoformans* species complex isolates were identified during this study (4/509, 0.8%). All four isolates were recovered from cultures of tree hollow swabs collected in the Mediterranean area, while there were no isolates recovered from bird excreta or from samples collected in the continental area. The isolate from Rijeka was found in the hospital area and three isolates from the island of Krk were found along the promenade beside the sea.

Comparisons of positive samples between the two sample types, between samples collected in the Mediterranean and continental area, as well as between those recovered from olive trees and other tree species were not statistically significant.

### 3.3. Molecular Characterization of C. neoformans Species Complex Isolates

The four *C. neoformans* species complex isolates were classified into two different molecular types, VNI (*n* = 3) and VNIV (*n* = 1), all with mating type α (Figure 2). The three VNI isolates originated from two locations, Rijeka and Krk Island, whereas the VNIV isolate was found on Krk Island. Three isolates were found in tree hollows of *O. europea*, while one VNI isolate was also found in a tree hollow of a *P. sylvestris*. The only isolate found in *P. sylvestris* tree hollows originated from the town of Rijeka and the isolates recovered from *O. europea* were found on Krk Island.

### 3.4. Antifungal Susceptibility of C. neoformans Species Complex Isolates

Results obtained by the ATB FUNGUS 3 method showed that all isolates were as susceptible as wild-type strains, with MIC values very similar for all antifungals tested. Antifungal susceptibility according to the CLSI broth microdilution method, performed for one VNI and one VNIV isolate, confirmed the ATB FUNGUS 3 results (Table 1).

## 4. Discussion

This is the first report showing the presence of *C. neoformans* species complex isolates from environmental sources in Croatia along with their antifungal susceptibility and molecular characterization, which were only available for clinical isolates so far [9]. However, studies had previously been conducted in an attempt to find *C. neoformans* species complex in the environment in this geographical area. An extensive environmental survey was carried out during 2012–2015, representing 12 countries with 6436 samples, which documented the distribution of *C. neoformans* and *C. gattii* species complexes around the Mediterranean basin. Croatia was represented with 18 environmental samples, but had no positive results [31]. Seasonal distribution cannot be the reason for not finding *C. neoformans* species complex in Croatia at the locations in the previous study because samples were collected each month by each of the collaborating groups. In that study, the peak of positive trees was observed during the spring. In our study, samples were also collected year-round, and all four isolates were found in May and June, which is in accordance with a previous study. However, the seasonal distribution cannot be analyzed because of the small number of positive samples [31].

In our study, *C. neoformans* species complex isolates were found only in the environmental from the Mediterranean part of Croatia. Their recovery from tree hollows of *O. europea* and *P. sylvestris* is in agreement with the aforementioned study investigating *C. neoformans* species’ complex distribution around Mediterranean basin and demonstrating the presence of this species in samples from trees belonging mainly to the genera *Eucalyptus*, *Olea*, *Ceratonia*, and *Pinus* in Cyprus, France, Greece, Italy, Libya, Portugal, Spain, and Turkey [30]. Leaves and flowers were not sampled because in the study by Cogliati et al. no isolates were recovered from these sites [31].

Tree hollows, as an important environmental niche for *C. neoformans* species complex, have been investigated since the 1990s [32]. The affinity of *Cryptococcus* spp. with this environment had been explained in previous studies by its ability to produce laccase enzyme. Aside from the production of melanin, laccase is involved in the degradation of lignin, a polymer related to cellulose and an integral part of virtually all higher plants that provides rigidity to woody tissue [33]. In our study, VNI isolates were recovered from two tree species, *O. europea* and *P. sylvestris,* while the VNIV isolate was only recovered from *O. europea*. In a previous study involving the Mediterranean area, *C. neoformans* var. *grubii* colonized 12 different tree genera, confirming its ability to adapt to different environments and distribute globally. In contrast, *C. neoformans* var. *neoformans* showed a preference for trees typical of the subcontinental climate, such as *Platanus, Prunus*, and *Quercus*, probably reflecting its ability to tolerate lower temperature better than *C. neoformans* var. *grubii.* However, the climatic zones have unclear boundaries, enabling isolates belonging to different varieties to share the same tree species as a niche [31,34].

Although the authors intended to collect both types of samples (swab samples of tree hollows and bird excreta in the soil beneath trees) from every location where samples were collected, bird excreta were difficult to find. The streets in towns were cleaned every day and samples were collected early in the morning; bird excreta were found only in two locations, Rovinj and Korčula. This may be a sign to other researchers investigating *C. neoformans* species complex in the environment that the improvement of sample collection techniques for this purpose is needed. The birds from which excreta were collected were not identified.

Bird excreta, if collected at two locations, could be easily positive as well, assuming that both the birds and the decaying trees are an important part of the life cycle of the same organism, *C. neoformans* species complex. In the recent study by Springer et al., the authors demonstrated the broad importance of plants (and plant debris) as an ecological niche and reservoirs of infectious propagules of cryptococci in the environment. Cryptococci can undergo saprobic filamentation, mating, and the production of spores on dead plant material, indicating the potential for the long-term association of cryptococci with plant [35]. Therefore, it may be possible that birds are responsible for spreading, while plants may help with maintaining, optimizing, and enhancing virulence factors during the environmental lifecycle of this opportunistic pathogen.

The distribution of molecular types VNI and VNIV in the environmental samples in Croatia was expected according to previous studies. VNI is the most prevalent molecular type in the Mediterranean basin, distributed from Portugal to Libya [5]. Molecular type VNIV, although less common than VNI, was already detected in the environment of Croatia’s neighboring countries, for example Italy [29,36]. In Greece, the VNIV is the most common molecular type, as well as in northern Turkey. Molecular types VNII, VNB, and VNIII were not detected in our study. So far, these types were detected in the Mediterranean basin from trees in Libya, Greece, and Turkey [29]. Although *C. gattii* species complex isolates were not found, their presence in Croatia, especially in the southern part of the country, cannot be excluded since this zone is characterized by a hot Mediterranean climate similar to that in the neighboring regions of southern Italy and Greece, where this pathogen has already been reported [29,36].

The molecular analysis of environmental isolates in Croatia demonstrated the presence of molecular types VNI and VNIV, which were already identified in clinical samples during previous studies. Molecular type VNIII has only been detected in clinical samples so far [5,9]. The impact of different molecular types on patient outcome in terms of clinical manifestations or attributable mortality rates is still unknown. Controversial results have been generated regarding the correlation between molecular types, virulence, and mortality rate [37,38].

The tested antifungal compounds amphotericin B, 5-flucytosine, fluconazole, itraconazole, and voriconazole demonstrated high in vitro activity against environmental *C. neoformans* species complex isolates. This is in accordance with the antifungal susceptibility results for clinical isolates in a previous study in which all tested antifungals also showed high in vitro activity against *C. neoformans* isolates [9]. Some studies demonstrated differences in antifungal susceptibility among molecular types, but the correlation between the molecular type and antifungal susceptibility is still an open issue [39,40].

This is the first report proving the presence of *C. neoformans* species complex in the environment of Croatia. The results of the study show the potential risk of exposure for inhabitants, especially on the Croatian coast, to certain molecular types, particularly VNI and VNIV, which can be expected in clinical cases of cryptococcosis. Although a limitation of this study is the small number of detected environmental isolates, their antifungal susceptibility pattern suggests that no strains with low susceptibility to the most common antifungals should be expected in patients with cryptococcosis and no guideline modifications are needed at the moment. Further investigation regarding the correlation of molecular types with clinical outcome and antifungal susceptibility are warranted.

## Figures and Tables

**Figure 1 jof-05-00099-f001:**
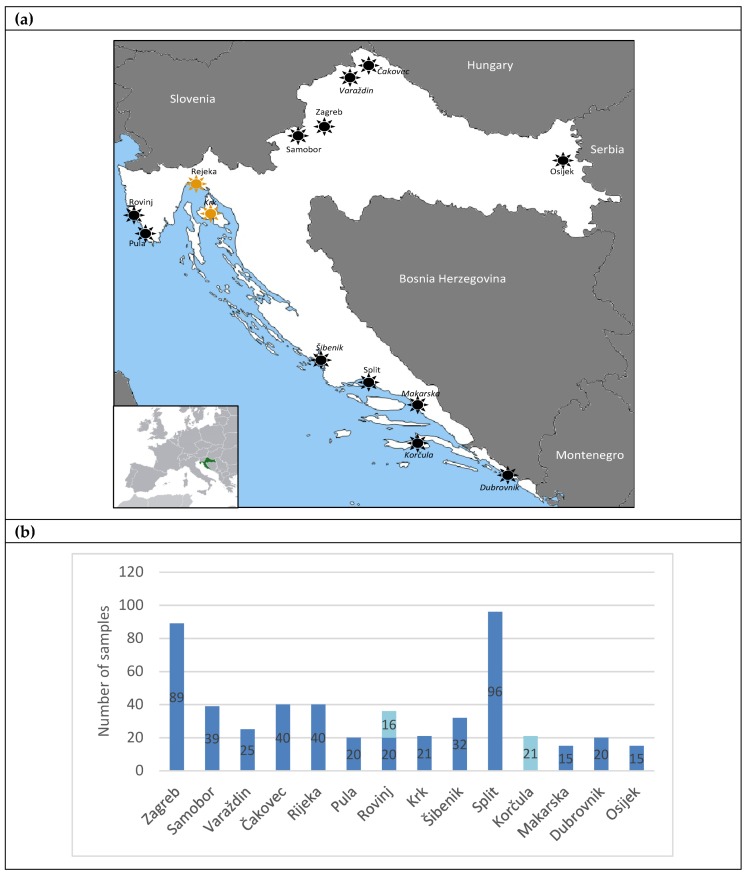
(**a**) The map represents the geographic distribution of the environmental samplings. Locations marked in yellow indicate the towns where positive samples were recovered. The inset shows the geographic position of Croatia in Europe. (**b**) The number and type of sample collected in each of the 14 sampling locations. Dark blue bars, samples from tree hollows; light blue bars, samples of bird excreta in the soil beneath the trees.

**Figure 2 jof-05-00099-f002:**
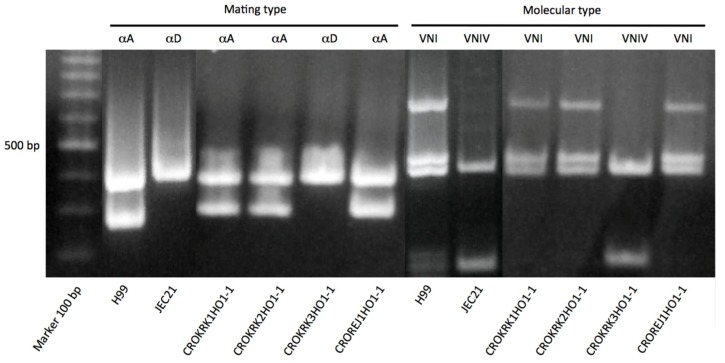
Multiplex PCR results showing mating type and molecular type identification of the four Croatian *Cryptococcus neoformans* species complex isolates. H99 and JEC21 represents the reference strains, VNI-αA and VNIV-αD, respectively. The 100 bp DNA ladder (Promega, Madison, WI, USA) was used as molecular DNA ladder. CROKRK = isolates from Krk Island; CROREJ = isolate from Rijeka.

**Table 1 jof-05-00099-t001:** Antifungal susceptibility of the four *C. neoformans* species complex isolates tested by ATB FUNGUS 3 and broth microdilution reported as minimal inhibitory concentration values (µg/mL).

Location	Molecular Type	5-FC (ECV = 8 µg/mL)		AMB (ECV = 0.5 µg/mL)		FLZ (ECV = 8 µg/mL)		ITZ (ECV = 0.25 µg/mL)		VOZ (ECV = 0.25 µg/mL)	
		ATBF3	CLSI	ATBF3	CLSI	ATBF3	CLSI	ATBF3	CLSI	ATBF3	CLSI
Krk	VNIV	4	-	0.5	0.25	1	4	0.125	-	0.06	0.25
Krk	VNI	4	-	0.5	-	1	-	0.125	-	0.06	-
Krk	VNI	4	-	0.5	-	1	-	0.125	-	0.06	-
Rijeka	VNI	4	-	0.5	0.25	2	4	0.125	-	0.06	0.125

5-FC = 5-fluorocytosine; AMB = amphotericin B; FLZ = fluconazole; ITZ = itraconazole; VOZ = voriconazole; ECV = epidemiological cutoff value; ATBF3 = ATB FUNGUS 3 method; CLSI = CLSI standard broth microdilution method; - = not tested.

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
