# Peer review of "First Isolation, Antifungal Susceptibility, and Molecular Characterization of *Cryptococcus neoformans* from the Environment in Croatia"

_jof, 2019, doi:10.3390/jof5040099_

Round 1
Reviewer 1 Report
Summary
This study set out to (a) investigate the presence of Cryptococcus neoformans and C. gattii in the various environmental sources in Croatia, and (b) determine their molecular types and antifungal susceptibility of isolated strains. The results showed that C. neoformans was present in tree hallows in 2 locations, no Cryptococcus was in bird excreta and no C. gattii was present in analyzed samples. Isolated Cryptococcus species were from two different molecular types; VNI (3 isolates) and VNIV (1 isolate).
Merits
This manuscript shows for the first time the presence of Cryptococcus neoformans in the environmental samples from Croatia. The authors did a good job of collecting and analyzing samples from all around Croatia (14 towns) and from various sources; tree hallows (46 different tree species) and bird excreta.
Questions/comments/suggestions
Why were bird excreta collected in only 2 locations? How did author determined where to collect only tree hallow samples or tree hallows and bird excreta? Only tree hallows were analyzed in the two locations that had positive neoformans samples. Do you think bird excreta from those locations will test positive for the presence of Cryptococcus? On the Lines 112-113 it is written that “Isolates were also tested for with CLSI”. However, in the results part (lines 189-190), only two isolates were tested by CLSI. I will suggest to be precise in the materials and methods that only 2 isolates were tested by CLSI. This is a question of curiosity. Where were the positive Cryptococcus samples found with the various parts tested (squares, parks, hospital areas, school playgrounds)? Here are some minor errors/typos Line 81. Is there a period missing before the word “Research”? Table 1. Add in the table legend the meaning of “-”. Is it not tested? Line 204. There is a red comma in the 6436 value that I think might be a typo. Reference 1. “Cryptococcus” should be in italic. Reference 9. “In vitro” should be in italic. Reference 25. “Cryptococcus neoformans” and “Cryptococcus gattii” should be in italic. Line 389. Separate “Transplantrecipients” into 2 words. Line 393. Separate “diversityand” into 2 words.
Author Response
Response to Reviewer 1 Comments
Point 1 and 2: Why were bird excreta collected in only two locations? How did author determined where to collect only tree hallow samples or tree hallows and bird excreta?
Response: When we were planning our study, our intention was to collect both type of samples (swab samples of tree hollows and bird excreta in the soil beneath trees) on every location where samples were collected. However, bird excreta were difficult to find. The streets in towns were cleaned every day and although we tried to look for samples early in the morning, we were able to collect this type of sample only in two locations of Rovinj and
Korčula. This maybe an information for other researchers investigating C. neoformans species complex in the environment that improvement of sample collection technique for this purpose should be improved. The paragraph referring to this issue was added to Discussion page 6, line 226).
Point 3: Only tree hallows were analysed in the two locations that had positive C. neoformans
samples. Do you think bird excreta from these locations will test positive for the presence of
Cryptococcus?
Response: Birds excreta, if collected at two locations, would be easily positive as well
assuming that both, the birds and the decay trees, are the important part of the life cycle of
the same organism, C. neoformans species complex. In the recent study by Springer et al.,
authors demonstrated the broad importance of plants (and plant debris) as the ecological
niche and reservoirs of infectious propagules of cryptococci in the environment. It can
undergo saprobic filamentation, mating and the production of spores on dead plant material,
implicating the potential for long-term association of cryptococci with plant. Therefore, it may
be possible that birds are responsible for spreading while plants may serve in maintaining,
optimizing and enhancing virulence factors during environmental lifecycle of this
opportunistic pathogen. The paragraph referring to this issue was added to Discussion page
6, line 226) and the reference by Springer et al. was added to References under number 33
(page 6, line 232).
Point 4: On the Lines 112-113 it is written that ‘’Isolates were also tested for with the CLSI’’.
However in the results part (line 189-190) only two isolates were tested by CLSI. I will suggest
to be precise in the materials and methods that only two isolates were tested by CLSI. This is
a question of curiosity.
Response: Antifungal susceptibility testing according to CLSI (Clinical Laboratory Standards
Institute) was done in 2019, two years after other investigations were already finished. For
that reason not all isolates (but two) were viable and available for this testing. Explanation
was added to Materials and methods (page 3, line 118).
Point 4: Where were the positive Cryptococcus samples found with the various parts tested
(squares, parks, hospital areas, school playground)?
Response: In Rijeka the positive sample was found in the hospital area, while in island Krk
they were found along the promenade beside the sea. This observation was added in Results
(page 5, line 176).
Point 5: Here are some minor errors/typss: line 81. Is there a period missing before the
word “Research”? Table 1. Add in the table legend the meaning of ’’-“. Is it not tested? Line
204. There is a red comma in the 6436 value that I think might be a typo. Reference
1.’’Cryptococcus’’ should be in italic. Reference 9.’’In vitro’’ should be in italic. Line 389.
Separate ‘’Transplantrecipinets’’ into 2 word. Line 393. Separate’’diversityand ‘’ into 2
words.
Response: Period was added before the word “Research” in line 81. Meaning of “-“ was
added in the table legend of Table 1. Comma was deleted in the 6436 value in the line 204
(now line 206). “Cryptococcus” is changed in italic in reference 1. “In vitro” is changed to italic
in reference 9. “Transplantrecipients” is separated in two words in line 389 (now line 408).
“Diversityand” is separated into two words in line 393 (now line 411).
Additional comment of the authors:
Page 2, line 79 – In this study we only investigated the presence, molecular typing and
susceptibility of environmental isolates of C. neoformans species complex. For that reason we
corrected the following sentence “The purpose of this study was to investigate the presence
of C. neoformans and C. gattii species complex isolates from environmental sources in
Croatia…” to “The purpose of this study was to investigate the presence of C. neoformans
species complex isolates from environmental sources in Croatia…”
Reviewer 2 Report
Pllana-Haydari and colleagues have identified the environmental presence of Cryptococcus spp., particularly species complex VNI and VNIV in Croatia. This is the first reported time that Cryptoccocus was found in the environment (particularly in tree hollows), and could attribute to recent cryptococcal infections. The manuscript would offer individuals and clinicians in the region the information on where Cryptococcus was identified and from what particular type of tree it was isolated, however the materials and methods could be further described.
It would be nice to know how trees were identified and if there's resources cited.
When were samples taken, the manuscript described from Cogliati et al in 2016 noted variation in tree colonization depending on the month. I wonder if you would note increases/decrease isolation based on time of year when the samples were taken. Would that also the reason why regions previously analyzed in the previous manuscript (towns of Rovinj, Pula, Osijek ) were unable to find Cryptococcus near Rejeka and Krk? It could be discussed further.
It would also be nice to offer rationale of why only tree hollows were sampled in trees, did you note less Cryptococcus in branches, leaves, flowers?
Was there a reason why there was less bird excreta tested than trees? Could the authors also identify any of the birds who's excreta was taken from?
Was the anti-fungal assay done using ATB Fungus 3, if so please incorporate in table 1. Was the analysis conducted in replicates if so please state it in the methodology.
In the introduction please include Samarasinghe and Xu 2018 recent publication in cryptococcal classification.
This this manuscript is rather short, the authors could show the verification of PCR identification in supplemental figure confirming the molecular and mating type.
Minor corrections:
Abstract last sentence, line 34 : add times following present.
Line 40 and 45. has double space at the beginning of the sentence
Line 81, insert period before Research
Materials and methods have double indentation in the beginning of the sentence
Line 89-91 may need rephrasing
large spacing from lines 164-169
Results have 3 indentations.
Author Response
Response to Reviewer 2 Comments
Point 1: It would be nice to know how trees were identified and if there’s resources cited?
Response: The trees were identified based on the shape, leaves, leave type and flower.
Point 2: When were samples taken, the manuscript described from Cogliati et al. in 2016
noted variation in tree colonization depending on the month. I wonder if you would note
increases/decreases isolation based on time of year when the samples were taken. Would
that also the reason why regions previously analysed in the previous manuscript (towns of
Rovinj, Pula, Osijek) were unable to find Cryptococcus near Rijeka and Krk? It could be
discussed further.
Response: Seasonal distribution cannot be the reason for not finding C. neoformans species
complex in Croatia at the locations in previous study because samples were collected
monthly, year round, by each of the collaborating groups. In that study peak of positive trees
was observed during spring. In our study samples were also collected year round and all four
isolates were found in May and June which is in concordance with study by Cogliate et al.
However, seasonal distribution cannot be analysed because of small number of positive
samples. The paragraph with this observation was added to Discussion together with cited
reference by Cogliati et al. (page 6, line 209).
Point 3: It would also be nice to offer rationale of why only tree hollows were sampled in
trees, did you note less Cryptococcus in branches, leaves, flower?
Response: Leaves and flowers were not sampled because in the study by Cogliati et al. no
isolates were recovered from these sites. The sentence was added in Duscussion (page 6,
line 221).
Point 4: Was there a reason why there was less bird excreta tested than tress? Could the
authors also identify any of the birds who’s excreta was taken from?
Response: When we were planning our study, our intention was to collect both type of
samples (swab samples of tree hollows and bird excreta in the soil beneath trees) on every
location where samples were collected. However, bird excreta were difficult to find. The
streets in towns were cleaned every day and although we tried to look for samples early in
the morning, we were able to collect this type of sample only in two locations of Rovinj and
Korčula. This maybe an information for other researchers investigating C. neoformans species
complex in the environment that improvement of sample collection technique for this
purpose should be improved. The birds from which excreta were collected were not
identified. The paragraph referring to this issue was added to Discussion page 6, line 236).
Point 5: Was the antifungal assay done using ATB Fungus 3, if so please incorporate in the
Table 1.
Response: Antifungal susceptibility of C. neoformans species complex isolates to flucytosine,
amphotericin B, fluconazole, itraconazole and voriconazole were determined by ATB FUNGUS
3 (BioMérieux, Marcy l'Etoile, France). Results are already incorporated in Table 1 in column
named “ATBF”. ATB FUNGUS was changed to ATB FUNGUS 3 in the title and legend of Table
1 and in the whole manuscript as well.
Point 6: Was the analysis conducted in replicates if so please state it in the methodology.
Response: Analysis was not done in replicates.
Point 7: In the introduction please include Samarasinghe and Xu 2018 recent publication in
cryptococcal classification.
Response: Publication by Samarasinghe and Xu 2018 was added in Introduction (page 2, line
64) and in References under number 16.
Point 8: This manuscript is rather short, the authors could show the verification of PCR
identification in supplemental figure confirming the molecular type and mating type.
Response:
Point 9: Minor corrections
- Abstract last sentence, line 34: add times following present
- Line 40 and 45, has double space at the beginning of the sentence
- Line 81 – insert period before Research
- Material and methods have double indentation in the beginning of the sentence
- Line 89-91 may need rephrasing
- Large spacing from lines 164-169
- Results have 3 identiations
Response: Times following present is added in Abstract last sentence. Double space is
corrected in Line 40 and 45 at the beginning of the sentence. Period is inserted before
Research in Line 8. Double identation is corrected in Material and methods in the beginning
of the sentence. Lines 89-91 are rephrased. Large spacing from lines 164-169 is corrected.
Three identations are corrected in Results.
Additional comment of the authors:
Page 2, line 79 – In this study we only investigated the presence, molecular typing and
susceptibility of environmental isolates of C. neoformans species complex. For that reason we
corrected the following sentence “The purpose of this study was to investigate the presence
of C. neoformans and C. gattii species complex isolates from environmental sources in
Croatia…” to “The purpose of this study was to investigate the presence of C. neoformans
species complex isolates from environmental sources in Croatia…”
Reviewer 3 Report
In this study, authors reported the first environmental isolates of Cryptococcus neoformans from Croatia, which confirms that C. neoformans is also present in this region. Among 509 samples collected from tree hollows and bird excreta in soil, four of them were C. neoformans. Molecular typing using multiplex PCR showed three of them are VNI strains and one as VNIV. They all showed normal antifungal sensitivity, indicting no drug resistance in these natural isolates. Overall, it is a survey report that is well written with positive data. It is nice to report the first environmental isolates in this country. However, only four positive isolates were identified, which may not enough to make any significant claim besides to show that C. neoformans VNI and VNIV are present in Croatia coast region, which should not be too surprising. It would really help if author could obtain and include some clinical isolates and/or historic data on clinical cases, to see whether there is any correlation between environmental isolates and clinical infections. As current form, a more specialized journal may be more appropriated for this report.
Specific comments:
Table 1. What does the “-“ indicates? Please state whether they are not tested or means something else. In addition to cite references (18, 27, 28), it may be necessary to generate a table to specify the primers used, and state what is the PCR conditions. Also a PCR result may be presented as the evidence of molecular types of each isolate. Line 176, “continental”
Author Response
Response to Reviewer 3 Comments
Point 1: It would really help if author could obtain and include some clinical isolates and/ or
historic data on clinical cases, to see whether there is any correlation between environmental
isolates and clinical infections. As current form, a more specialized journal may be more
appropriated for this report.
Response: Similarity regarding molecular types and antifungal susceptibility between
environmental and clinical isolates in Croatia is already mentioned in the Discussion
paragraph of the manuscript. (page 7, line 275 and 281) .
Point 2: Table1. What does ‘’-‘’ indicates?
Response: Meaning of “-“ is “not tested” and was added in the table legend of Table 1.
Point 3: In addition to cite reference (18, 27, 28) it may be necessary to generate a table to
specify the primers used and state what is the PCR conditions. Also PCR result may be
presented as the evidence of molecular types of each isolated.
Response: The multiplex PCR methods used to identify mating type and molecular types of
theC. neoformans species complex isolates was described more than 15 years ago and were
used to type hundreds of strains worldwide. The references cited in the text contain the
details of primers sequences and PCR conditions. We have added a new figure (figure 2) and
the corresponding legend showing multiplex PCR results for molecular type and mating type
identification of Croatian isolates.
Point 4: Line 176,’’continental’’.
Response: In Line 176 “contintental” was changed to “continental”.
Additional comment of the authors:
Page 2, line 79 – In this study we only investigated the presence, molecular typing and
susceptibility of environmental isolates ofC. neoformans species complex. For that reason we
corrected the following sentence “The purpose of this study was to investigate the presence
ofC. neoformans and C. gattii species complex isolates from environmental sources in
Croatia…” to “The purpose of this study was to investigate the presence of C. neoformans
species complex isolates from environmental sources in Croatia…”